# Turbulent mixing and heat fluxes under lake ice: the role of seiche oscillations

Georgiy Kirillin[1], Ilya Aslamov[2], Matti Leppäranta[3], and Elisa Lindgren[4]

[1]Deaprtment of Ecohydrology, Lebiniz-Institute of Freshwater Ecology and Inland Fisheries (IGB), Berlin, Germany
[2]Limnological Institute, Siberian Branch of Russian Academy of Science (LIN SB RAS) Irkutsk, Russia
[3]Institute of Atmospheric and Earth Sciences, University of Helsinki, Helsinki, Finland
[4]Department of Physics, University of Helsinki, Helsinki, Finland

**Correspondence:** Georgiy Kirillin (kirillin@igb-berlin.de)

**Abstract.** We performed a field study on mixing and vertical heat transport under ice cover of an Arctic lake. Mixing intensities were estimated from small-scale oscillations of water temperature and turbulent kinetic energy dissipation rates derived from current velocity fluctuations. Well-developed turbulent conditions prevailed in the stably stratified interfacial layer separating the ice base from the warmer deep waters. The source of turbulent mixing was identified as whole-lake (barotropic) oscillations of the water body driven by strong wind events over the ice surface. We derive a scaling of ice-water heat flux based on dissipative Kolmogorov scales and successfully tested against measured dissipation rates and under-ice temperature gradients. The results discard the conventional assumption of nearly conductive heat transport within the stratified under-ice layer and suggest contribution of the basal heat flux into the melt of ice cover is higher than commonly assumed. Decline of the seasonal ice cover in the Arctic is currently gaining recognition as a major indicator of climate change. The heat transfer at the ice-water interface remains the least studied among the mechanisms governing the growth and melting of seasonal ice. The outcomes of the study find application in heat budget of seasonal ice on inland and coastal waters.

## 1 Introduction

Seasonal formation of ice cover is an essential feature of the hydrological regime of temperate and polar climatic zones. Freshwater lakes are, in this regard, a special class of hydrological objects, both in terms of thermal and hydrodynamic processes that control the formation and melting of ice, and from the point of view of the impact of the ice regime of the global freshwater budget. The majority of world lakes is concentrated in the northern temperate and boreal environments between 40°N and 80°N and have the potential to freeze seasonally (Kirillin et al., 2012). Seasonally ice-covered lake systems—Lake Baikal, Laurentian Great Lakes, European Great Lakes Ladoga and Onego, lake systems of Fennoscandia and Northern Canada—accumulate the bulk of the world's surface freshwater. Their seasonal ice regime determines climatic balance of precipitation and evaporation, as well as the ecosystem state and the water quality of lakes themselves. The lakes of high latitudes remain

less studied than temperate lakes, while arctic regions are reported to have the strongest air temperature increase (Screen and Simmonds, 2010). Steady trend to the shortening of the ice season on lakes during the last 100-150 years was reported in a large number of recent studies (e.g. Benson et al., 2012; Bernhardt et al., 2012; Karetnikov et al., 2017) and was attributed to global climate warming. Although long-term observations of changes in lake ice phenology (seasonal appearance and duration of the

ice cover) are distinct climate change indicators, they do not provide any quantitative information on the processes underlying these changes. Therefore, physics of seasonal ice cover on lakes is gaining recognition as an essential part of climate change, playing a key role in greenhouse gas emission and the global carbon budget. Estimation of the consequences of phenological changes on water resources requires quantification of the physical mechanisms that control the formation and melting of ice. The heat and mass transfer at the ice-water interface is the least studied among these mechanisms.

The ice-water interface in freshwater lakes is distinguished by two specific features: Lack of permanent average water flow (in contrast to rivers) and hydrodynamic instability due to density anomaly of freshwater (in contrast to oceans). As a result, the most energetic mixing process in lakes is the convective mixing in the upper water column due to solar radiation penetrating the ice cover (Kirillin et al., 2012). Since the temperature at the ice-water interface is fixed at the freezing point of freshwater, a strongly stratified interfacial layer (IL) is formed beneath the ice-water interface with temperatures decreasing upward from

that of the convective mixed layer (CL) to 0 °C. The strong stratification suppresses turbulent transport, so that conduction is suggested to be the major heat transfer mechanism within the IL. For typical thickness of IL of $\leq 1$ m, the conductive heat flux is $\leq 1$ W m$^{-2}$. The values reported in the literature are close to this estimation, but tend to exceed it at later stages of the ice cover melting: (Bengtsson and Svensson, 1996) estimated the heat flux from water to ice in a number of small Swedish lakes as ranging within 1-2 W m$^{-2}$ in January-February and increasing to 5-7 W m$^{-2}$ in March-April. Jakkila et al. (2009) reported

values in ice-covered Lake Pääjärvi of 5 W m$^{-2}$ in winter and 12 W m$^{-2}$ in late spring. Heat fluxes of 7-29 W m$^{-2}$ were measured by Leppäranta et al. (2010) during the final stage of ice melting in Lake Vendyurskoye.

The assumption of conductive, turbulence-free IL does not hold true in large lakes, where under-ice water circulation is strong enough to produce long-lasting shear turbulence at the ice base: Aslamov et al. (2014, 2017) reported a strong, up to an order of magnitude, variability in the ice-water heat flux in Lake Baikal at synoptic time scales of several days and referred it

to variations of the large-scale surface currents, as a part of geostrophic circulation in Lake Baikal.Aslamov et al. (2014) also demonstrated that the resulting increase of the heat supply from water to the ice cover may cancel the ice growth and produce bulk melting even at atmospheric temperatures below the freezing point of water and upward heat flux at the ice surface. Any quantitative information on mixing conditions in the IL of small [as compared to the Rossby radius, see (Kirillin et al., 2012)] ice-covered lakes was missing up to date. The strong diurnal variability of the convection driven by solar heating (Petrov and

Sutyrin, 1984) and the evidence of basin-scale standing waves under ice (Malm et al., 1998) suggest that the upward heat flow to the ice base is non-stationary and affected by shear production of turbulence even in lakes free from large-scale horizontal circulation.

The present study is aimed to quantify the contribution of turbulent mixing to the ice-water heat exchange, as a crucial mechanism behind the seasonal ice cover formation. For this purpose, we performed a field campaign on an arctic lake with a

focus on mixing in the ice boundary layer and ice-water heat exchange. Based on direct measurements of small-scale turbulent

velocity and temperature fluctuations in the IL, we demonstrate that the ice boundary layer is far from quiet, with shear turbulence generated by transient events of seiches (standing basin-scale waves). We investigate the effect of shear turbulence on the upward heat transport and, consequently, the ice cover melting. In view of the new findings, the validity is discussed of the commonly used approaches to parameterization of the ice-water heat exchange.

5     While the process of ice melt is also affected by a complex transformation of the solar radiation within the ice cover as well as by the surface radiative balance and the turbulent fluxes at the ice surface, We intentionally limited the scope of the study to the heat transport in the water column and its influence on the ice-water heat exchange. By this, we carefully considered the processes governing formation of temperature gradients at the ice base, including convective heating of the ice-covered water column and dynamics of the temperature profile in the stratified interfacial layer under ice driven by diurnal variations 10  in convection and by wind-driven ice oscillations. In turn, the heat budget at the ice surface and within the ice cover are left to be a subject of a parallel study by Leppäranta et al. (2018).

## 2   Methods

### 2.1   Study site

The field campaign was performed on 21-27 May 2014 in Kilpisjärvi (Fig. 1)—a "mid-size" (surface area: 37.1 $km^2$) lake 15  located in the Scandinavian Mountain Chain in northwest Finland (69° 03′N 20° 50′E, 473 m above sea level). The average and maximum depths of the lake are 19.5 m and 57 m, respectively. The average ice-on and ice-off dates are 9 November and 18 June (mean value over 1952-2015), the earliest recorded ice-on date was October 21 and the earliest and the latest breakup date were June 2 and July 1, respectively (data of station of the Finnish Environment Institute (Korhonen, 2006). In summer, surface water temperatures arrive at 10–15 °C in mid-August. On average, the maximum annual ice thickness is reached in 20  mid-April, amounting at 89 cm (the range over 1952-2015 was 77–114 cm). Weather data (standard observations, including air temperature at 2 m height and wind speed at 10 m height) were adopted from the Kilpisjärvi Biological Station and Enontekiö Kilpisjärvi Kyläkeskus (EKK). The latter station was founded in Kilpisjärvi village in 1962 by the Finnish Meteorological Institute. The annual mean air temperature is between $-4$ °C and $-1$ °C, with mean monthly air temperatures below 0 °C from October to April. Snow thickness on ground in March–April is on average 90 cm. The snow thickness on the lake ice tends to 25  be lower than on the surrounding land, partly due to snow removal by winds, partly due to transformation of snow to snow-ice (Leppäranta et al. 2017), with an annual maximum of $37 \pm 9.1$ cm.

### 2.2   Vertical heat transport in the water column

The background information on the vertical temperature distribution was collected by a thermistor chain installed close to the lake center (69° 2′ 3.09″N, 20° 46′ 45.87″E, depth 32 m). The chain was equipped with 9 temperature loggers TR-1060 (RBR, 30  Canada, declared accuracy 0.002° C, sampling period 10.0 s) at heights 31.25 27.00 25.00, 20.80, 18.65, 14.40, 8.25, 6.20, 4.15 m above the bottom, and a temperature-pressure logger RBR-DUO (RBR, Canada, temperature accuracy 0.002 °C, pressure

accuracy 0.05% of 50 dbar, sampling period 2.0 s) at 1.00 m above the bottom. Another temperature-pressure logger recorded atmospheric pressure at the lake shore for correction of the water level records. Several casts of vertical profiles of temperature and conductivity were performed during the study using the RINKO CTD profiler (JFE Advantech, Japan, temperature accuracy 0.01 °C, conductivity accuracy 0.01 mS cm$^{-1}$, sampling rate 10 Hz). Ice and snow thickness were measured during the CTD casts with an accuracy of $\pm 0.5$ cm.

Data on photosynthetically active radiation (PAR) were collected by three compact PAR loggers DEFI (Alec Electronics and JFE Advantech, Japan), one on the ice surface, one frozen into the ice cover, and one in the water column at 0.3 m beneath the ice cover. All sensors recorded planar quantum irradiance with 10 minute intervals. The spectrum of solar short-wave (wavelengths range 200-2500 nm) radiation is strongly modified by lake waters: clear water (or ice) quickly absorbs the long-wave (infrared) part of the spectrum and yellow substance absorbs the short-wave (ultraviolet) part. As a result, at $< 1$ m depth, $> 95\%$ of the penetrated radiation falls within the PAR spectral range of 400-700 nm (see e.g. Jerlov, 1976). In humic brown-water lakes, like Kilpisjärvi, the equivalence between PAR and total solar radiation is even closer (e.g. Leppäranta et al., 2010). Hence, we adopted the measured PAR values as characteristic of the total downward short-wave radiation flux. Transformation between the measured quantum irradiance $I_q$ [ μmol s$^{-1}$ m$^{-2}$] and the net downward short-wave radiation $Q_I$ [ W m$^{-2}$] was obtained in the atmosphere by the relation $I_q/Q_I = 4.60$ μmol J$^{-1}$ (see Leppäranta et al., 2010). In water and ice the coefficient depends on the color and turbidity of water. In Finnish and Estonian lakes the transformation coefficient has been found to vary between 4.8 μmol J$^{-1}$ and 5.5 μmol J$^{-1}$, larger values referring to more turbid waters (Reinart et al., 1998). For the clear-water Lake Kilpisjärvi the coefficient was chosen as 4.8 μmol J$^{-1}$. The accuracy of the PAR sensors is $\pm 2$ W m$^{-2}$ in air and $\pm 3$ W m$^{-2}$ in water or ice. For a comprehensive description of PAR measurements see Leppäranta et al. (2018). Light extinction within the water column was estimated from the vertical profiles of PAR taken during the previous field campaign in 2013 (Kirillin et al., 2015). The profiles were measured in 2013 with a spherical quantum PAR-sensor LI-193SA (LiCor, USA) attached to a CTD-90M profiler (Sea&Sun Technology, Germany). The light extinction coefficient $\gamma$ and the radiation value at the ice-water interface $Q_I0$ were determined from a one-band exponential approximation of the short-wave radiation profile $Q_I(z)$ in the water column,

$$Q_I(z) = Q_I0 \exp(-\gamma z) \tag{1}$$

The vertical turbulent heat flux within the bulk of the water column $Q(z,t)$ as function of time $t$ and depth $z$ was estimated from temperatures measured by the thermistor chain $T(z,t)$ using the "flux-gradient method" which adopts the 1-dimensional equation of heat transfer, with horizontal advection neglected:

$$C_p\rho \frac{\partial T(z,t)}{\partial t} = -\frac{\partial Q(z,t)}{\partial z} - \frac{\partial Q_I(z,t)}{\partial z}, \tag{2}$$

where $C_p\rho \approx 4.18 \cdot 10^6$ J K$^{-1}$ m$^{-3}$ is the product of the water heat capacity and density. The solar radiation flux $Q_I(z,t)$ was recovered from PAR measurements and Eq. (1). Integration of Eq. (2) from a reference depth $H$, usually chosen close to the

lake bottom, to a depth $z$ yields the turbulent flux $Q(z,t)$ as

$$Q(z,t) = Q(H,t) + Q_I(H,t) - Q_I(z,t) - C_p\rho \int\limits_{H}^{z} \frac{\partial T(\zeta,t)}{\partial t} d\zeta \qquad (3)$$

Assumed that, at the end of the ice-covered period, the bottom sediment is in thermal equilibrium with the deep water layers, i.e. the vertical turbulent flux $Q(H) \approx 0$, Eq. (3) was solved numerically using finite differences for differentiation and trapezoid method for integration.

## 2.3 Measurements in the ice-water boundary layer

Measurements of the turbulent pulsations of current velocities were performed using a pulse-coherent high-resolution acoustic Doppler profiler HR-Aquadopp (Nortek AS, Norway). The profiler was attached to a thin plate with positive buoyancy and deployed alongside the ice cover base, with the acoustic head oriented looking downwards at right angle to the instrument. The profiler registered three components of the current velocity vector at sampling rate 0.5 Hz with vertical resolution of 0.02 m.

The HR-Aquadopp data were used for estimation of the dissipation rate $\varepsilon$ of the turbulent kinetic energy (TKE). The procedure of $\varepsilon$ estimation generally followed that from the previous study (McGinnis et al., 2015): The velocity structure function

$$D(z,r) = \left\langle (v(z) - v(z+r))^2 \right\rangle \qquad (4)$$

was calculated from the HR-Aquadopp data and was used for estimation of the TKE dissipation rate, as described by Wiles et al. (2006). Here, $v(z)$ is the along-beam velocity fluctuation at distance $z$, $r$ is the depth range of the dissipation estimation, angle brackets denote time averaging. Three estimations of the velocity fluctuations were determined by extracting the mean velocity value for the averaging periods of 10, 20, and 30 min. Three values of the maximum estimation range for the velocity correlation $r = 0.4$, 0.5, and 0.6 m were tested, covering the range recommended by Wiles et al. (2006) for weakly stratified turbulence. The TKE dissipation rate was estimated by fitting the equation

$$C_v^{-3} D(z,r)^{3/2} = r + Noise. \qquad (5)$$

Here, the constant $C_v = 3^{1/3}$ (see e.g. Lien and D'Asaro, 2002). The fitting constant $Noise$ representing the average effect of the acoustic noise on the velocity fluctuations was used for the goodness-of-fit check, using the condition

$$Noise > \langle C_v^{-3} D^{3/2} \rangle. \qquad (6)$$

The measurements, for which Eq. (6) was valid, were abandoned as noisy. Further quality check was performed by comparison of the 27 arrays of the TKE dissipation rate calculated from the three HR-Aquadopp beams with three different values of r and three averaging periods. The discrepancy between the different estimations did not exceed 10% (not shown). The estimations based on the averaging time of 20 min were adopted for the further analysis averaged over the three beam estimations. The lower value of $r = 0.4$ m was adopted to avoid occasional influence on the results of the increased instrumental noise at higher distances from the acoustic head.

To measure the vertical distribution and short-term variability of water temperature within the strongly stratified ice-water boundary layer a custom measurement setup was used. It consisted of a temperature registration platform with 12 fast-response temperature loggers T-Solo fast16 (RBR, Canada, accuracy 0.002 °C, time constant 0.1 s, sampling frequency 16 Hz) equipped with stainless steel sledges on the upper side and attached to the remote operated vehicle VideoRay Pro4 (VideoRay, USA). The distance from the ice base to the uppermost sensor was 5 cm; other loggers were distributed vertically at distances of 2 cm between sensors to the depth of 27 cm from the ice base. The platform was able to slide along the ice bottom up 150 m away from the deployment site to locations undisturbed by instrument installation. Two loggers of the downward solar radiation were additionally attached to the platform, but malfunctioned during the deployment. Series of 24 to 48-hour long deployments were performed with continuous temperature registration at sampling frequency of 2 Hz.

## 2.4  2-dimensional model of standing barotropic waves

The variations in the temperature and turbulent mixing within the ice-water boundary layer were analyzed for their relationship to the standing barotropic waves (seiches) by using spectral analysis and modeling of the free oscillations of the lake. Spectral densities were determined using the standard periodogram method. The free oscillation modes of Kilpisärvi were estimated from numerical solution of the eigenvalue problem

$$\lambda_k^2 \Psi = -c^2 \left( \frac{\partial^2 \Psi}{\partial \mathrm{x}^2} + \frac{\partial^2 \Psi}{\partial \mathrm{y}^2} \right) \tag{7}$$

on the geometry of the lake in the horizontal coordinates $x$ and $y$. Here, $c = (gH)^{1/2}$ is the long wave speed (celerity) based on flat-bottom approximation of the lake bathymetry with the mean depth $H$, $\lambda k$ is the eigenfrequency and $\Psi$ is the eigenfunction of the harmonic level oscillations $\zeta$ in time $t$, defined as

$$\zeta(t) = \sum_{\mathrm{k}=1}^{\infty} \Psi \exp(i\lambda_k t) \tag{8}$$

Eq. (7) was solved on a triangular mesh approximating the Kilpisärvi geometry (Fig. 1C) using the finite element solver of COMSOL Multiphysics software. The lake geometry was provided by National Land Survey of Finland.

## 2.5  Approximate solutions of the vertical heat transport equation

As mentioned above, the ice-water interface in small lakes is commonly assumed to be turbulence free. Assuming the heat transport within IL is steady-state ($\partial Q \partial t^{-1} = 0$) and non-turbulent ($Q(z) = -\kappa \partial T \partial z^{-1}$), a direct solution of the steady-state heat transport equation (2) turns to

$$\kappa \frac{d^2 T}{dz^2} - \frac{dI}{dz} = 0 \tag{9}$$

Subject to boundary conditions (Fig. 2)

$$T(0) = 0; T(\delta) = T_m,$$

Eq. (9) yields the solution (Barnes and Hobbie, 1960)

$$T(z) = \frac{z}{\delta} T_m + \frac{1}{\kappa} \left( \int_0^z I(\zeta) d\zeta - \frac{z}{\delta} \int_0^\delta I(\zeta) d\zeta \right) \tag{10}$$

Here, $\delta$ is the IL thickness, $\kappa$ is the heat conduction coefficient, $I = (C_p \rho)^{-1} Q_I$ is the kinematic flux of the short-wave solar radiation, $T_m$ is the temperature of the convectively-mixed layer beneath IL. The IL thickness $\delta$ can in turn be found by applying the condition of smooth temperature transition between IL and the thermally-homogeneous CL beneath, i.e.

$$\left. \frac{\partial T}{\partial z} \right|_{z=\delta} = 0 \tag{11}$$

or,

$$\kappa T_m + \delta I(\delta) - \int_0^\delta I(\zeta) d\zeta = 0 \tag{12}$$

The latter equation can be solved with respect to $\delta$ if the function $I(z)$ is known. For a single-band exponential decay of the short-wave solar radiation within the water column (1), the solution (12) turns to

$$\gamma \kappa T_m + I_0 ([1 + \gamma \delta] e^{-\gamma \delta} - 1) = 0 \tag{13}$$

which equation is solved numerically with respect to $\delta$. As it follows from Eq. (13), $\delta$ depends on the CL temperature $T_m$, the radiation flux $I_0$ and the water transparency $\gamma$: the more turbid is the lake, the stronger is the radiation, the higher is $T_m$ and the thinner is IL. The water-ice heat flux is determined directly from the solution (10) as

$$Q_{cw} = -\kappa \left. \frac{\partial T}{\partial z} \right|_{z=0} = -\frac{1}{\delta} \left( \kappa T_m - \int_0^\delta I(\zeta) d\zeta \right) - I(0) = I(\delta) - I(0) \tag{14}$$

i.e. the upward heat flux equals to the solar radiation absorbed within the IL in the steady-state non-turbulent conditions and no temperature gradient (zero heat flux) at the IL bottom.

Another simple model, particularly useful for the analysis below, is that of non-penetrative growth of the CL on the background of stable density stratification in the quiescent layer (QL) beneath. The approximation was first introduced by Zubov (1945) and was later named "encroachment" (Carson and Smith, 1975) in contrast to the "entrainment" regime, which involves entrainment of a denser fluid from QL to CL. If the well-mixed convective layer develops on top of the QL by increase of the mixed layer temperature $T_m$, without changing the quasi-linear temperature gradient $\Gamma$ in the QL, the rate $dh\,dt^{-1}$ of the CL deepening and the heating rate are related as

$$\frac{dh}{dt} = \Gamma^{-1} \frac{dT_m}{dt} \tag{15}$$

which suggests zero turbulent (heat) buoyancy flux at the CL bottom. Together with the previous assumption of the zero flux at the CL-IL boundary, the CL development can be estimated from data on solar radiation flux:

$$\frac{dh}{dt} = \Gamma^{-1} \frac{I(\delta) - I(h)}{h - \delta} \tag{16}$$

This regime is strictly never realized in natural convection (Carson and Smith, 1975) but can be closely approached, at early stages of convection development (Carson, 1973).

## 3 Results

### 3.1 Weather and ice conditions

During the observations period of 22-27 May, the air temperature did not change much, with the mean value of 2.15 °C ± 1.75, minimum of -2.00 °C and maximum of 5.30 °C. Low winds of 2-4 m s$^{-1}$ prevailed most of the period, with a moderate wind increase up to 9 m s$^{-1}$ on 22 May and a short event of strong winds >12 m s$^{-1}$ on 24-25 May (Fig. 3). Conditioned by mostly cloudy weather and a thick ice-cover with more than 50% of snow-ice, the short-wave radiation flux right beneath the ice cover amounted at daily means of 10-20 W m$^{-2}$, with daily peak values 35–70 W m$^{-2}$ (Fig. 3), which made up 6-7% of the solar radiation flux at the ice surface.

The ice was snow-free (snow thickness < 1 cm). The ice thickness at the beginning of the field campaign was close to the seasonal maximum values: mean 97 cm, min 82 cm, max 126 cm, based on 29 measurements from a cross-section along a short axis of the lake. 52 cm of the ice cover consisted of "white ice" (snow-ice), or frozen slush on top of the congelation ice. From 20 May to 26 May the ice thickness reduced to 73 cm (minimum 61 cm, maximum 80 cm, based on 37 measurements from a cross-section along a short axis of the lake).

for the period under investigation, the ice surface temperature remained close to the freezing point of 0 °C, which also implies that the entire ice cover had nearly constant temperature of 0 °C. The heat budget of the ice cover is dominated in this case by the heat exchange at the ice bottom and by the absorption of solar radiation within the ice cover, whereas the conductive heat transport across the ice cover is small. The situation is characteristic of the early stage of the ice cover melt and was considered in detail by Aslamov et al. (2014). Since half of the ice sheet was snow-ice (white ice), its low transmittance limited the transfer of sunlight to water and also the ice experienced internal deterioration from absorption of solar radiation. At the upper boundary the heat balance was positive and caused surface melting. The surface heat balance was dominated by the solar and long-wave radiation balance, turbulent fluxes being small. Upper surface melting and internal melting accounted for 1–2 cm per day. Further details on the structure, optical properties, and heat budget of the ice cover are presented elsewhere (Leppäranta et al., 2018).

### 3.2 Temperature structure of the water column

The vertical temperature distribution had three-layer structure, typical for the spring convection in ice-covered lakes due to solar heating (Fig. 4, cf. also Fig. 2): the well-mixed convective layer (CL) had initial thickness $h_m \approx 5.5$ m and temperature $T_m = 0.8$ °C, adjacent to a 1.5-m thick stratified interfacial layer (IL) on top with temperature decrease to the freezing point at the ice-water interface. The quiescent layer (QL) beneath the CL was nearly linearly stratified with downward temperature increase of $\Gamma \approx 0.04$ K m$^{-1}$ (hereinafter Kelvins used for temperature gradients, while absolute temperatures are given in degrees Celsius for convenience). Solar radiation penetrating the ice cover caused increase of temperature in the CL at $1.8 \cdot 10^{-2}$ °C day$^{-1}$ and increase of its thickness at 0.33 m day$^{-1}$.

In contrast to observations at later stages of the convection development in ice-covered lakes (see e.g. Mironov et al., 2002), there was no appreciable temperature/buoyancy jump at the bottom of the CL: the well-mixed layer $h$ developed on top of

the QL by increase of the mixed layer temperature $T_m$, without changing the quasi-linear temperature gradient $\Gamma$ in the QL, as described by the "encroachment" model (Eq. 15). At the top of the CL, in turn, convection produced an increase of the temperature gradient across the IL from 0.58 K m$^{-1}$ on 20 May to 1.74 K m$^{-1}$ on 26 May that corresponds to an increase of the conductive heat flux across the IL from 0.33 to 0.99 W m$^{-2}$ according to the model of Barnes and Hobbie (Eq. 14).

## 3.3 Heat budget within the water column

The light attenuation within the water column was estimated based on 24 PAR profiles taken in May 2013 as $\gamma = 0.30 \pm 0.05$ m$^{-1}$. We adopted this estimation for the 2014 campaign, in suggestion of low year-to-year variability of $\gamma$ for the same season. The resulting vertical profiles of the radiative flux (Fig. 5CD) were used for estimation of the vertical turbulent heat transport (black lines in Fig. 5AB) by vertical integration of the heat equation (Eq. 3). During daytime, the turbulent fluxes were generally directed downwards, increased across the convective mixed layer towards the lake surface to $\sim$10 W m$^{-2}$, and dropped to almost zero within the stratified surface layer at the ice base: an indication of the turbulence damping by stratification in the surface layer. Exact values of the turbulent fluxes in the vicinity of the ice-water interface were difficult to estimate by the heat budget method due to coarse vertical resolution. No appreciable negative fluxes existed at the bottom of the convective mixed layer ($\sim$7 m depth), in agreement with the absence of entrainment layer in the temperature profiles. Also, the low but non-zero turbulent fluxes below the convective mixed layer (at 10-25 m water depths) may have resulted from the method constrains, but could be produced by natural processes, such as horizontal heat advection of vertical mixing by internal waves. During nighttime, turbulent transport changed its direction to upward. The fluxes were generally below 10 W m$^{-2}$, except the night of 24-25 May, when strong flux of $\sim$18 W m$^{-2}$ developed near the ice base, decreasing downwards across the mixed layer.

## 3.4 Dynamics of stratified interfacial layer (IL)

The stratified IL occupied the up to 2-m thick layer under the ice base, separating the CL from the ice base. High-resolution temperature observations during 24-26 May in the IL demonstrated periodic fluctuations with 24 hour period and amplitude of 0.2-0.3 K, related to the subsurface heating by the penetrating solar radiation. On the background of the diurnal cycle, higher-frequency temperature oscillations persisted during the first half of the observation period (Fig. 6a). Irregular high-amplitude temperature fluctuations lasted for the first 5-6 hours (24 May, 13:00-19:00 in Fig. 6a), and were replaced later by harmonic quasi-sinusoidal oscillations (24 May 19:00 – 25 May 19:00 in Fig. 6a) decaying within $\approx$24 hours. Concurrently with the strong temperature fluctuations, the TKE dissipation rate $\varepsilon$ increased by an order of magnitude $> 10^{-8}$ W kg$^{-1}$ in the entire IL, as estimated from the velocity fluctuations (Fig. 6b). The burst in the turbulent mixing decayed within 5-6 hours to the background values at the limit of the detectable turbulence level of $\varepsilon \approx 10^{-10}$ - $10^{-9}$ W kg$^{-1}$. The burst in TKE dissipation and temperature fluctuations coincided with the wind increase on 24-25 May (Fig. 3).

The dataset on the TKE dissipation rate covered a longer period than the temperatures in the IL, and demonstrated a close connection between wind speeds over the lake and the turbulence intensity under the ice cover: apart from the strong wind burst on 24-25 May, the winds of $\sim$10 m s$^{-1}$ on 22 May increased $\varepsilon$ to $\sim 10^{-8}$ W kg$^{-1}$ (cf. solid line and wind bars in Fig. 7. The

whole series of 22-27 May suggests that wind has an appreciable effect on mixing under ice at speeds $> 10$ m s$^{-1}$, otherwise, the mixing intensity in the IL remains low, with $\varepsilon \approx 10^{-9}$ W kg$^{-1}$. Weaker increases of $\varepsilon$ up to $10^{-8.5}$ W kg$^{-1}$ took place at quasi-diurnal intervals and were connected apparently to the periodical increase of radiation-driven convection in the CL underneath (cf. temperature variations in Fig. 6A).

The vertical temperature gradients $\partial T \partial z^{-1}$ under ice (line with circles in Fig. 7) demonstrated strong correlation with dynamics of wind and $\varepsilon$ during the same period, following the decay of the mixing intensity from its peak on 24 May. Vertical conductive heat flux, calculated from these gradients as $-\rho c_P \varkappa \partial T \partial z^{-1}$, remained as low as 1.0-1.5 W m$^{-2}$. However, the high dissipation rates under ice (Fig. 6B) suggest the heat transport within this layer remains mainly turbulent and exceeds the conductive one. The estimations of the water-ice heat flux from the model of Barnes and Hobbie (1960, see Eq. (14)) are approximately 2 times higher ($2.0 - 3.0$ W m$^{-2}$), but do not reflect the temporal variability connected to $\varepsilon$, because they relay on the assumption of a steady-state IL.

Spectral analysis of the under-ice temperature and pressure records demonstrated strongly periodic character of water motions. The spectral energy of water level (pressure) variations (Fig. 8A) was concentrated at several peak frequencies, fairly corresponding to the eigenfrequencies from the two-dimensional seiche model (Eq. (7)). However, the three different stages distinguished in the subsurface temperature time series (cf. Fig. 6A)——"wind burst" on 24 May, "free oscillations" on 25 May, and "convection only" on 26 May—reveal rather different spectral patterns: During the "wind burst", most of energy still resided at the low-frequency "gravest seiche mode" oscillations (triangles in Fig. 8A). The energy peaks in temperature oscillations during the same period (Fig. 8B) were relatively low, suggesting that the vertical motions in the stratified IL were only weakly periodic. The "free oscillations" stage was characterized by concentration of motions around the frequencies of the second and higher seiche modes, i.e. vertical motions with the maximum amplitudes away from the lake shores. During the "pure convection" stage the spectral energy in the range short-term oscillations 1-100 cph was essentially zero (not shown), i.e. at relatively low winds of $< 5$ m s$^{-1}$ no seiche oscillations were produced under the ice cover.

## 4   Discussion

Information on the energetics of ice cover formation and decay in polar lakes remains limited to date. The present results reveal several important aspects of the physics behind seasonal ice cover melting. One part of the results, such as comparably low water-ice heat flux following from conventional estimation methods and development of the convective mixed layer driven by solar radiation, add new details to previously known information. The other results—production of strong turbulent mixing by wind-driven fluctuations of ice cover, seiche oscillations under ice, and the direct connection of the TKE dissipation rate within the IL to the water-ice heat flux—demonstrate quantitatively new facets of the seasonal ice cover dynamics.

Despite the seasonal maximum thickness of the ice cover, the amount of solar radiation penetrating the ice cover (Fig. 3) was sufficient to produce a convective mixed layer (CL) under ice at the initial stage of the melting period (Fig. 4). The vertical structure of CL was distinguished by the absence of a strong temperature (density) gradient at its bottom, characteristic of the developed convection on later stages of the ice-covered period, when CL actively entrains into the stratified quiescent layer

beneath (Mironov et al., 2002; Kirillin et al., 2012). Such a form of the CL development corresponds to the "encroachment" regime, with zero buoyancy flux at the bottom of CL. However, the real situation was strongly non-stationary, driven by the diurnal variability in the solar radiation, so that the entrainment flux should also vary at diurnal time scales. Farmer (1975), following Deardorff et al. (1974), noted that even small non-zero flux at the CL base significantly affects $\mathrm{d}h\,\mathrm{d}t^{-1}$. In the observed case, the encroachment-like CL development was facilitated by the diurnal character of the negative buoyancy production and by relatively weak solar heating: while during daytime solar radiation produced downward turbulent fluxes in the upper water column, the nighttime fluxes changed its direction, contributing primarily to distribution of added heat across the CL and to upward heat release into the IL (Fig. 5AB). Apparently, entrainment of the warm water at the base of CL took place only during the day, with subsequent nighttime erosion of the entrainment layer. Therefore, care should be taken when applying the encroachment model to the under-ice convection, even at the early stage of the convection development.

The estimations of the water-ice heat flux based on the measured temperature gradients within the IL and those following from simplified steady-state solution of the heat transport equation (14) did not exceed 2-3 W m$^{-2}$. The conductive flux estimations following from Eq. (14) provide about two times higher values than calculations based on the directly measured temperature gradients within IL. This fact suggests on one hand that the temperature gradients closer to the ice bottom are much higher than those measured at 5-7 cm under the ice. On the other hand, high rates of the TKE dissipation within the IL and its strongly non-stationary dynamics suggest that real heat exchange between the water column and the ice cover is appreciably stronger than predicted by the steady-state model of Barnes and Hobbie (1960). The difference between the mean radiation flux penetrating the ice ($\approx 15.8$ W m$^{-2}$, Fig. 3) and the water column heating ($\approx 6.9$ W m$^{-2}$, Fig. 4, Eq. 16) is close to 10 W m$^{-2}$, which value can be assumed as a "true" upward heat flux, averaged over the observations period. Hence, conventional methods of the flux estimation essentially—up to an order of magnitude—underestimate the real values by assuming nearly conductive conditions in the IL.

Spectral analysis of ice cover fluctuations revealed basin-scale standing waves (seiches) at the lake surface, as the major mechanism of the wind energy transfer through the ice cover to the under-ice water motions. As observed earlier (Bengtsson and Svensson, 1996; Malm et al., 1998; Baehr and DeGrandpre, 2002) and supported by modeling efforts (Sturova, 2007; Zyryanov, 2011), frequencies of standing waves are not significantly affected by the presence of the ice cover. The small discrepancies between the modeled and observed seiche frequencies in our case can be referred to limitation of the flat-bottom representation of the lake morphometry in the model. Our results, however, demonstrate an effect of the ice cover on the model pattern of seiches: the initial oscillations of the lake surface during the wind gust are characterized by the dominating first mode, i.e. the level variations are the strongest near the lake shores. During the subsequent "free seiching" the first mode quickly decays, in favor of higher modes—those having maximum amplitudes in the open parts of the lake. The latter are retained in a lake for a day. We interpret this redistribution of oscillations towards higher modes as a result of the damping effect of the ice cover fastened at the lake shores. Apart from a better understanding of the mechanism of the wind energy transport through the ice cover, the finding can have important implications for horizontal mass and heat transport under ice.

Our observations on the dissipation rates of the turbulent energy in the IL provide new important insights into the ice-water interactions. In particular, variations in mixing conditions take place on synoptic time scales and are related to wind events over

the ice cover. Wind-produced oscillations increase the energetics of mixing by an order of magnitude. As the ice gets thinner, and the stratification in the IL gets stronger, the role of the wind-induced mixing in the upward heat transport from the CL to the ice base is expected to significantly increase. The effect of the shear mixing under ice on the vertical heat transport is akin to the shear turbulence produced by the geostrophic circulation in ice-covered Lake Baikal (Aslamov et al., 2014, 2017),

with one important difference: a strong lake-wide circulation under ice takes place only in very large lakes, while production of shear turbulence by fluctuations of the ice cover is expected to develop in the majority of ice-covered lakes. Aslamov et al. (2014, 2017) reported heat fluxes of up to 50 W m$^{-2}$ at the ice base of Lake Baikal, associated with $\varepsilon = O(10^{-8})$ W kg$^{-1}$. We observed similar dissipation rates, hence similar flux magnitudes can be expected in small lakes during strong wind events, accelerating significantly ice melting.

The apparent relationship between the TKE dissipation rate and the upward heat flux suggests that $\varepsilon$ can be used as a directly measurable scale of ice-water heat flux $Q_{iw}$. The following scaling considerations can be applied for this purpose. The effect of turbulent mixing on the temperature gradient $dT\,dz^{-1}$ within the diffusive layer, and, consequently, on $Q_{iw}$ is twofold: (i) it affects the temperature at the outer boundary of $\delta$T, and (ii) it changes the thickness of $\delta T$ itself, thinning the diffusive layer by increasing turbulence outside. The former effect is relatively slow, since the typical temperature variations in the CL take place

at diurnal and longer time scales; the variations of $\delta T$ are in turn stronger, and can amount at an order of magnitude depending on mixing conditions. The thickness of the viscous sub-layer may be assumed proportional to size of the smallest turbulent eddies, or to the Kolmogorov's length scale (Monin and Yaglom, 1971),

$$\delta_\nu \propto L_\varepsilon \propto v^{3/4}\varepsilon^{-1/4} \tag{17}$$

where $\nu \approx 10^{-6}$ m$^2$ s$^{-1}$ is the molecular viscosity of water. Then, the heat flux at the ice base is expressed as

$$Q_{iw} = K\Delta T; \quad K \propto \mathrm{Pr}^n(\nu\varepsilon)^{1/4} \quad [\mathrm{m\,s}^{-1}] \tag{18}$$

where $\mathrm{Pr}^n$ reflects the relationship between diffusive and viscous microscales via the Prandtl number ($\mathrm{Pr} \approx 10$ for water), $K$ is the heat transfer coefficient, and $\Delta T$ is the temperature difference across the diffusive layer. The scaling (17) suggests that the temperature gradient in the close vicinity of the ice base roughly scales with the TKE dissipation rate as

$$\frac{dT}{dz} \propto \frac{\Delta T}{\delta} \propto \varepsilon^{1/4}. \tag{19}$$

Indeed, the scaling (19) works perfectly well in the whole range of the observed variability of $\varepsilon$, (Fig. 9), except for the later "quiet" period, when no oscillatory motions were present, and the TKE was produced only in the lower part of the IL by convection. Hence, our observational data confirm scaling of the water-ice heat flux (Eq. 18) by means of the Kolmogorov microscales, as long as turbulence is produced by velocity shear at the ice base. This important outcome of our rough scaling analysis opens a direct way for a general parameterization of the boundary heat (and mass) flux at the ice base as a function of

the boundary layer turbulence. Such a parameterization would be applicable to any shear-driven turbulent heat flow, like that produced by large-scale currents under ice in large lakes (Aslamov et al., 2014) and ice-covered seas (Peterson et al., 2017). Direct calculation of the water-ice heat flux from $\varepsilon$ has several advantages compared with the traditional bulk parameterizations

of the heat flux (McPhee, 1992): the TKE dissipation rate is a prognostic variable in many turbulence models and can be directly implemented for boundary fluxes parameterizations. Besides, a significant progress has been made during the last decades in methods of the turbulence dissipation measurements that allow long-term registration of $\varepsilon$ in situ and using this data for quantification of boundary-layer processes hardly measurable by other methods. Direct application of Eq. (18) to observational data requires determining of proportionality constants in Eq. (19). The latter would involve additional assumptions about the temperature and $\varepsilon$ profiles outside of the viscous sublayer $\delta_\nu$ and is subject of field experiments in different ice-covered environments, in particular, at various Reynolds and Richardson numbers in the IL.

## 5 Conclusions

The results presented above demonstrate the crucial role played by strong wind events in the heat transport across the strongly stratified water layer under seasonal ice. The basin-scale barotropic oscillations have been unambiguously identified as the mechanism of the kinetic energy transfer from wind to the under-ice turbulence. Such oscillations are inherent features of small lakes (as well as semi-enclosed seas and bays). A general conclusion following from our analysis is: an assumption on existence of a turbulence-free conductive layer under ice does not hold true even in relatively small ice-covered lakes. Hence, the previously reported values of the basal heat flux contribution to the seasonal ice melt were very probably underestimated. Another important outcome of the study is the scaling of the apparent under-ice heat flux against the Kolmogorov velocity scale based on the TKE dissipation rate. This result applies to any turbulent under-ice boundary layer—whether marine or freshwater—independently of the source of the TKE. A quantitative application of the scaling to parameterization of the ice-water heat exchange requires however exact establishing of the proportionality constant between the Komogorov length scale and the thickness of the viscous sub-layer, which can be obtained by fine-scale field measurements, laboratory experiments, or eddy-resolving numerical simulations.

*Data availability.* The datasets on lake water temperatures and TKE dissipation rates used in the analysis can be downloaded from http://www.flake.igb-berlin.de/LacunaData (Kirillin et al., 2017). Unprocessed high-frequency data on velocity and temperature fluctuations in the ice boundary layer are available from the first author by request. Meteorological data are courtesy of Finnish Meteorological Institute (https://en.ilmatieteenlaitos.fi/open-data)

*Competing interests.* The authors declare no competing interests

*Acknowledgements.* The research is part of the research project "IceBound" funded by the German Science Foundation (DFG project KI 853/11-1). The field campaign was performed in frames of the EU FP7 Program "International Network for Terrestrial Research and Monitoring in the Arctic (INTERACT)" via transnational access project "Lake circulation during polar night in Arctic (LACUNA)". Data anal-

ysis was carried out under LIN SB RAS state assignment No 0345–2016–0008. EL was supported by the Nordic Center of Excellence Cryosphere–atmosphere interactions in a changing Arctic climate (CRAICC). Data on wind velocity are provided by the Finnish Meteorological Institute. We thank the personnel of the Kilpisjärvi Biological Station and the rest of our field team for their invaluable help with our field campaigns. In particular, we are indebted to Antero Järvinen, the Station Director, for his continued support.

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

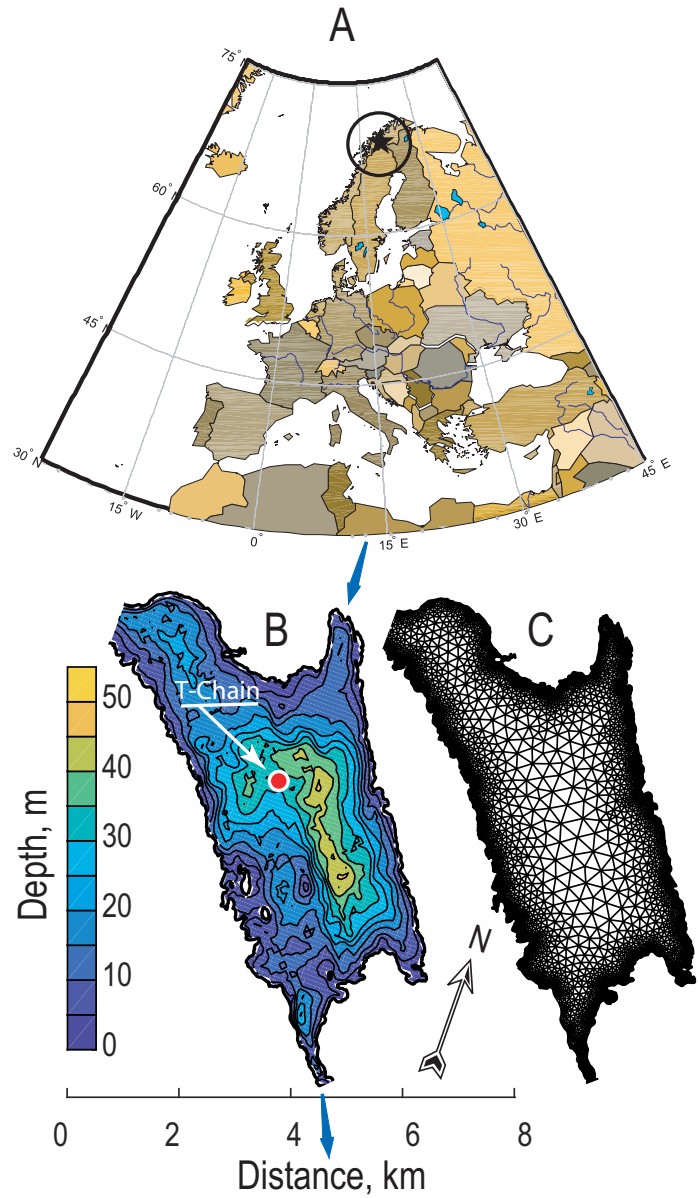

**Figure 1.** (a) Geographical location of the study site, (b) bathymetric map with the position of the thermistor chain, and (c) the grid used in the model of free oscillations. Blue arrows show the river flow direction.

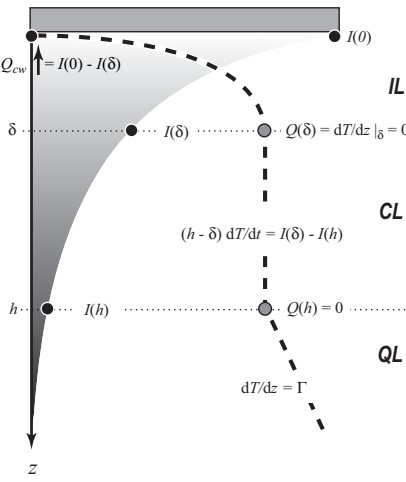

**Figure 2.** Schematic of temperature structure of the under-ice water column illustrating the major assumptions under the models (10)-(14) and (15)-(16)

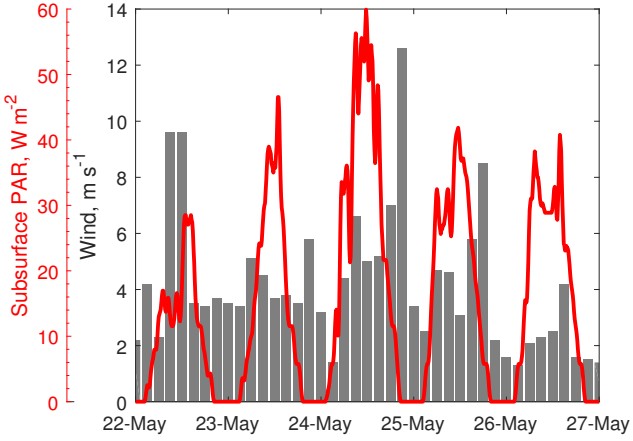

**Figure 3.** Wind (bars) and under-ice radiation (solid line) during the observations period

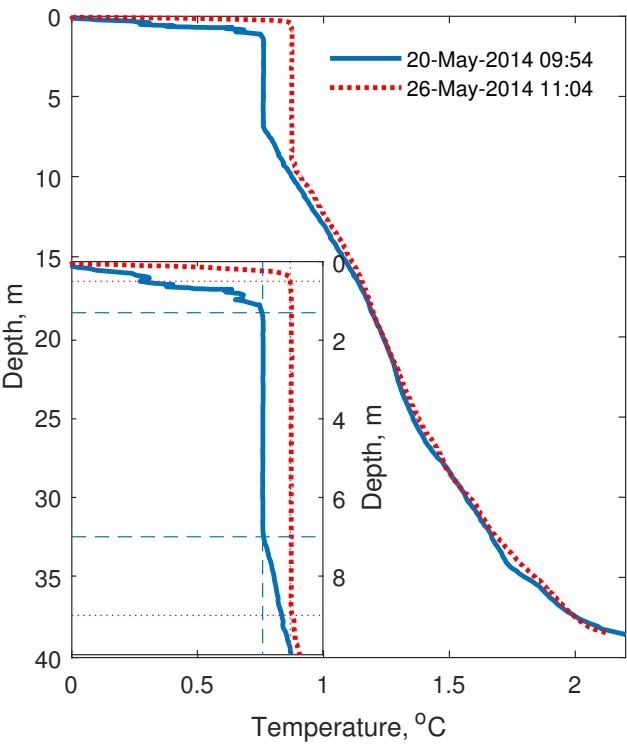

**Figure 4.** Temperature profiles at the beginning (solid line) and at the end (dotted line) of the field study. The inlet shows a zoomed portion with CL and temperature gradients at its boundaries

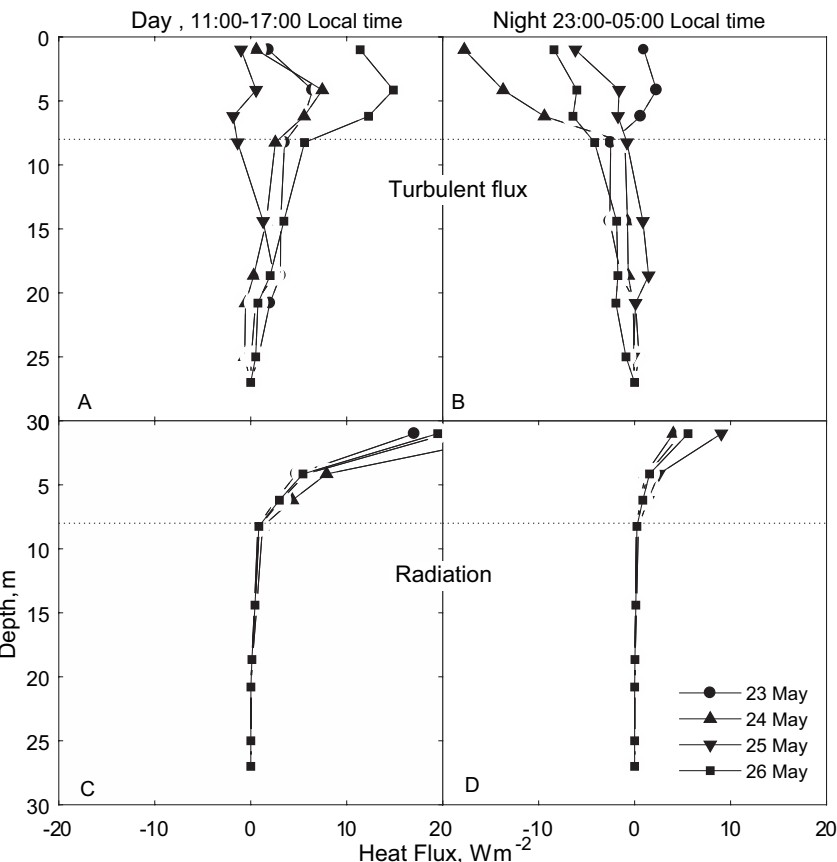

**Figure 5.** Vertical turbulent (A, B) and radiation (C, D) heat fluxes in the bulk of the water column. Panels A, C show the daytime averages, Panels B, D correspond to the nighttime values

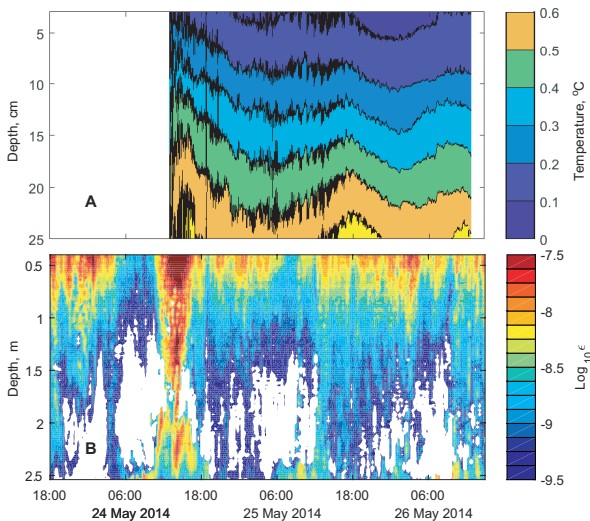

**Figure 6.** (A) Temperature fluctuations in the IL and (B) TKE dissipation rates in the upper water column. In white areas of Panel B no developed turbulence could be detected, because no inertial subrange could not be identified according to Eq. (5)

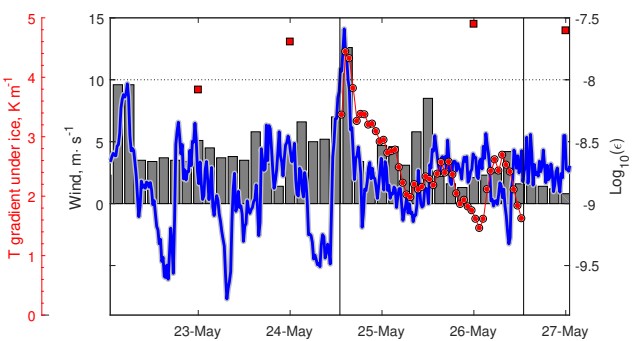

**Figure 7.** Vertically averaged TKE dissipation rate under ice (solid line) vs. Wind speed over the lake (bars) and vertical temperature gradient from the 2 uppermost temperature loggers under ice, at 5 and 7 cm under the ice bottom (line with symbols). Single squares show the mean daily vertical temperature gradient under ice according to the steady-state model of Barnes and Hobbie (1960), see Eq. (14). Note the different vertical axes for temperature, wind, and $\varepsilon$.

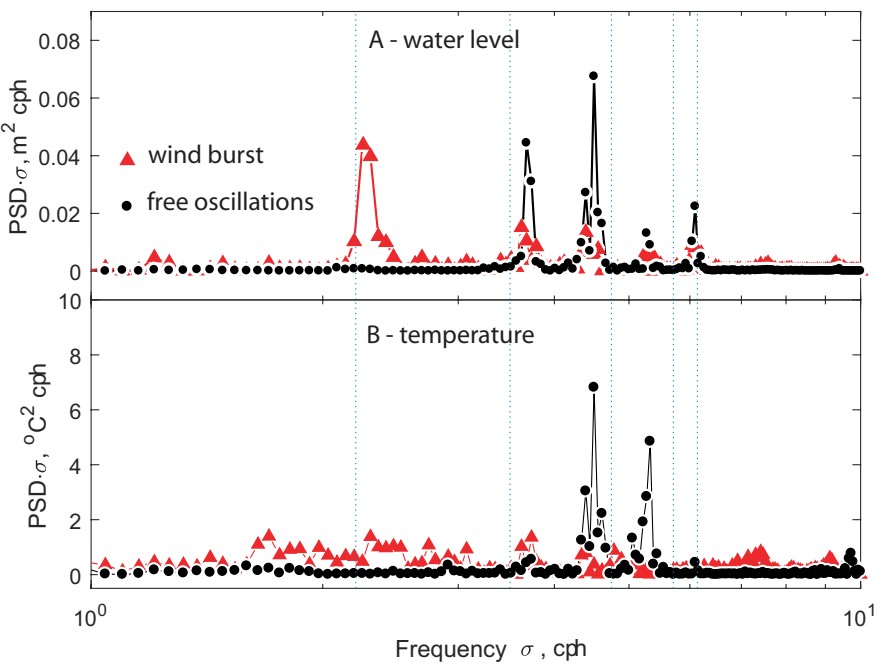

**Figure 8.** Power spectral densities for (A) water level variations and (B) temperatures in the IL for the "wind burst" 24 May 2014 12:00-24:00 (red triangles) and "free oscillations" 25 May 2014 03:00-15:00 (black circles). The vertical dotted lines mark the modeled frequencies of free oscillations in (flat-bottom) Kilpisjärvi

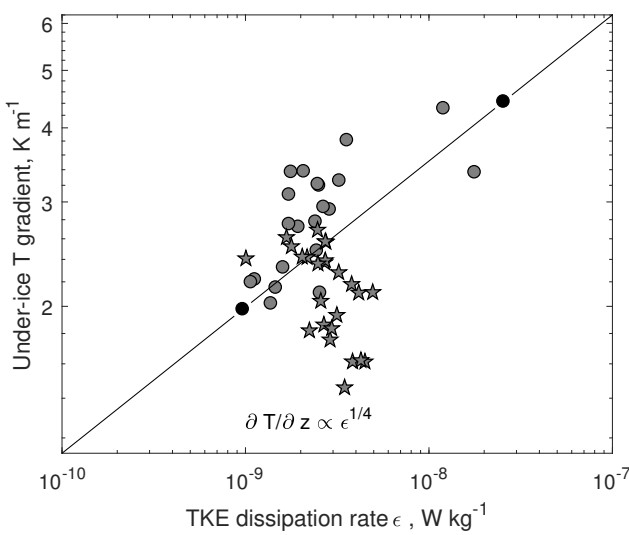

**Figure 9.** Relationship between the TKE dissipation rate and the temperature gradient in the uppermost part of the IL covered by the measurements (5-7 cm under ice). Instead of a least-square approximation, a simple line is drawn connecting the minimum and the maximum of $\varepsilon$ (full black circles): The slope of the line is 0.251. Stars mark the measurements during the period of "no wind-driven oscillations" (26 May 2014 00:00-12:00)