# Peer review of "Turbulent mixing and heat fluxes under lake ice: the role of seiche oscillations"

_Hydrology and Earth System Sciences, 2018_

## Referee Comment (RC1) · Anonymous Referee #1 · 1 Sep 2018

This paper proposes an extensive review and research on determining the turbulent mixing and heat fluxes under lake ice by a series of experiments and theoretical mechanics calculation. The article's experimental workload is very rich and the research contents are meaningful. However, there are some obvious technical and language description mistakes in this article. Therefore, I think this article is suitable for publication after some minor modification.

Comments and Suggestions for the Author(s)

Technical Advice: ïĄň For conducting some advances on the reading this article, please increase a flow chart of the test this paper at the end of section 2. Then, please add symbol list and calculation parameter table in this paper. Therefore, the readers can understand the process of experimentation. ïĄň In the paper, theoretical and simu-

lation calculations were carried out for the heat flow problem of fluids under lake ice. Some basic theories of fluid mechanics and thermodynamics were used in the calculation of the article. However, whether the overall fluid-solid coupling characteristics are taken into consideration during the temperature-dependent process of the ice body? That is, from the perspective of fluid-solid coupling thermodynamics, the influence of temperature on the mixed state of ice water and the effect of ice-water mixing state on the ambient temperature are estimated. ïĄň In this paper, the author divided the ice sheet into different layers on the base of depth. Whether or not the bulk structure of the ice material is considered to have a thermal effect on the different layers? From the ice material itself, ice crystals, bubbles, impurities, etc. have an important influence. Considering the ice layer, the air flow between the layers (although very small amount) and the water flow (mainly between ice and water) seem to have a certain influence on the temperature change. ïĄň Fig. 1 has shown the geographical distribution and depth distribution. It is great. However, ice materials are not artificial materials, and their formation has great natural laws and meanings. Among them, the flow direction of water and the flow speed of water have a very important influence. So, I suggest the author add hydrogeological map and water temperature change map of this lake in this paper.

Details: ïĄő Figures Please unify the line style annotations in the text. For instance, please modify Fig. 5 for using one group of line labeling for A/B/C/D. Please modify Fig. 8 for adding the group of line labeling for line red and black in the Fig. but not in the Fig. title. Please add symbol distinction in Fig.9.

English: The English of this paper is native. However, please split some long sentences into a few short sentences.

---

## Referee Comment (RC2) · Anonymous Referee #2 · 5 Nov 2018

The manuscript entitled "Turbulent mixing and heat fluxes under lake ice: the role of seiche oscillations" from Georgiy Kirillin, Ilya Aslamov, Matti Leppäranta, and Elisa Lindgren presents original results on energy transfers in lake water under ice cover. The wording is very clear and the English much better than I can judge. The paper is well structured and increases the knowledge on the mechanism of the transfer of wind kinetic energy to under-ice turbulence and its importance to the seasonal ice melt. The conclusions of the work may be very useful to improve the parameterizations of the boundary heat flux at the ice base, which are relevant to the sciences of hydrology and Earth System. So in my opinion the paper can be published after some minor improvements.

line 14: (Fig. 2): It is the first reference for a figure - it should be figure 1
line 15: The following sentence has no verb: "For typical thickness of IL of ≤ 1 m, the conductive heat flux amounts at ≤ 1 W m−2." Please rewrite

Page 3 – Section 2.1
The current Figure 1 should be referred to in this section.
If available, it would be useful to add more information on under-ice water climatology

In page 4, $I$ is defined as net PAR irradiance, but in page 6 $I$ is defined as the kinematic flux of the short-wave solar radiation. Please be consistent. In adition explain what you mean by short-wave solar radiation and which the relation between these and the PAR radiation. Please indicate the spectral bands of the definitions.
line 27: "sampling rate 0.5 Hz". This frequency is enough of to take in account all the relevant scales of the turbulence? Please comment.

line 10: It is not clear how *Noise* is computed?
line 11 "to discard corresponding values with subsequent interpolation between the neighboring values". I do not understand this sentence. Please explain better.
line 14: "The estimations based on the averaging time of 20 min and r = 0.4 m were adopted for the further analysis 15 averaged over the three beam estimations.". Why?
line 16: "For registration of". Maybe is better to write "To access" or "To measure"

line 24: "warmer is the water". In the IL, right?

line 4: $QL$ was not previously defined.
line 17 – 18: The level at which air temperature and wind were measured should be indicated.
line 18: s-1 → s$^{-1}$
line 23: The sentence "The ice was snow-free (snow thickness < 1 cm)" seems to be in contradiction to the information in line 20: "a thick ice-cover with more than 50% of snow-ice"
line 24: "a cross-section". The same cross-section used before?

Page 8:
line 14: "suggestion". Are there any evidence/reference for this suggestion?
line 16: To integrate Eq. 3 the radiative profiles are not enough. How have you compute the other terms?
line 28: "The stratified IL occupied the 2-m thick layer". According to Figure 4 the IL is thinner than 2-m

Page 9:
Line1: "the TKE dissipation rate ε increased". Increased relatively to what? Maybe the sentence should be rewritten beginning with the "background" values...
Line 5: "The dataset on the TKE dissipation rate covered a longer period than the temperatures in the IL". How, as TKE dissipation rate is computed from the temperatures?
line 14: " had no effect". Based on Fig. 7, it seems that there are some effect.

Label of figure 7: The scales in which each series is represented should also be indicated in the label.

---

## Author Comment (AC1) · 26 Nov 2018

We thank the reviewer for positive evaluation of our study and for the valuable comments aimed at improvement of the paper readability. We agree to the major remarks of the Reviewer and will incorporate the suggested amendments in the manuscript, unless they appear superfluous and a compromise between detailing and conciseness should be preferred.

The content-related comments/questions are discussed below:

**Q**: How were the temperature effects on the ice-water heat transport accounted for? What is the effect of the ice structure on the heat transport across the ice cover?

**A**: This fully legitimate comment points at an apparent (and intentional) gap in our study, regarding the heat budget closure of the entire air-ice-water system. Temperature conditions at the two boundaries—the air-ice and the ice-water interface—are indeed crucial for the heat budget of the ice cover. We, however, intentionally limited the scope of the present paper to the heat transport in the water column and its influence on the ice-water heat exchange. By this, we carefully considered the processes governing formation of temperature gradients at the ice base, including convective heating of the ice-covered water column (Section 3.2) and dynamics of the temperature profile in the stratified interfacial layer under ice driven by diurnal variations in convection and by wind-driven ice oscillations (Section 3.4). In turn, the heat budget at the ice surface and within the ice cover are left to be a subject of a parallel study by Leppäranta et al. (2018). There are certain physically justified reasons for this separation. First, the newly discovered mechanism of oscillatory produced turbulence and its effect on the ice-water heat flux required a dedicated analysis. Second, the physics of ice melt allows intuitive differentiation of processes above and beneath the ice base in a straightforward way, as described below.

The heat exchange at the upper ice boundary is dominated by the air temperatures, which varied during the observations period in the narrow range from $-1\,°C$ to $+2\,°C$ (see Section 3.1). Hence, for the period under investigation, the ice surface temperature remained close to the freezing point of $0\,°C$, which also implies that *the entire ice cover had nearly constant temperature of* $0\,°C$. Slightly positive air temperatures also imply stable conditions in the ice-atmosphere boundary layer, which additionally restrict the surface heat exchange between the atmosphere and the ice cover. The heat budget of the ice cover is driven in this case by the heat exchange at the ice bottom and by the absorption of solar radiation within the ice cover, *whereas the conductive heat transport accross the ice cover is small*. The situation is characteristic of the early stage of the ice cover melt and was considered in detail by Aslamov et al. (2014). A distinctive feature of small polar lakes, also mentioned by the Reviewer, is the inhomogeneous vertical structure of the ice cover. While, as mentioned above, the con-
ductive heat flux during the melting phase is weakly affected by the ice structure due to homogeneous temperature distribution within the ice cover, the crystal orientation, impurities, and air bubbles can play a critical role in absorption of the solar radiation by ice. This fact provides a tentative explanation for the relatively high ice melting rates during the observation period, and is considered in detail by Leppäranta et al. (2018). The structure of the ice has a major impact to the melting via the optical properties, albedo and transmittance. In 2014 half of the ice sheet was snow-ice (white ice), and its low transmittance limited the transfer of sunlight to water and also the ice experienced internal deterioration from absorption of solar radiation. At the upper boundary the heat balance was positive and caused surface melting. The surface heat balance was dominated by the solar + longwave radiation balance, turbulent fluxes being small. Upper surface melting and internal melting accounted for 1–2 cm per day.

To provide the reader with an appropriate context, a modified version of the above paragraph will be added to the revised paper.

**Q**: What is the influence of the under-ice water flow on the ice-water heat exchange?

**A**: We directly measured the flow rates in the under-ice boundary layer and did not observe any mean flow except the oscillatory water motions due to barotropic seiches (Section 3.4, Figs. 6 and 8). As we demonstrated in a previous study based on data from the same lake (Kirillin et al., 2015), the lake-wide under-ice circulation can indeed be strong under ice at the concluding stage of the ice-covered period, when lateral density gradients form by partial opening of an ice-covered lake. River inflows can also potentially contribute to the under-ice flow, but their effect is often localized along the flow path (Kirillin et al., 2012). The oscillating flows, as discussed in this study, dominate the turbulence production throughout the ice-covered period in small (in terms of the Rossby radius) lakes, whereas density currents are the major turbulence producers in large lakes (Aslamov et al., 2014).

Comments on the minor remarks:

- For the sake of conciseness, we respectfully decline the suggestion on adding of a list of symbols. The paper is not heavily loaded with mathematical derivations, and all symbols are defined at their first appearance. Adding an extra list would be repetitive and would require additional printing costs.

- To suggestion on adding a hydrogeological map: While we agree that the river inflow may potentially affect the ice cover growth/decay, the groundwater generally appears to have a minor influence on the ice cover regime of polar lakes. To avoid overloading of Figure 1 with auxiliary information, we will add the direction of in- and outflow in Kilpisjärvi to the figure.

- The rest of suggestions referred to the figure legends will be thankfully incorporated into the revised version of the paper.

**References**

Aslamov, I., V. Kozlov, G. Kirillin, I. Mizandrontsev, K. Kucher, M. Makarov, A. Y. Gornov, and N. Granin
2014. Ice–water heat exchange during ice growth in lake baikal. *Journal of Great Lakes Research*, 40(3):599–607.
Kirillin, G., A. Forrest, K. Graves, A. Fischer, C. Engelhardt, and B. Laval
2015. Axisymmetric circulation driven by marginal heating in ice-covered lakes. *Geophysical Research Letters*, 42(8):2893–2900.
Kirillin, G., M. Leppäranta, A. Terzhevik, N. Granin, J. Bernhardt, C. Engelhardt, T. Efremova, S. Golosov, N. Palshin, P. Sherstyankin, et al.
2012. Physics of seasonally ice-covered lakes: a review. *Aquatic sciences*, 74(4):659–682.
Leppäranta, M., E. Lindgren, G. Kirillin, and L. Wen
2018. Ice cover decay in Lake Kilpisjärvi in Arctic tundra: Solar radiation and heat budget of the ice cover. submitted to Aquatic Sciences.

---

## Author Comment (AC2) · 26 Nov 2018

We thank the reviewer for positive evaluation of our study and for the valuable comments aimed at improvement of the paper readability. We agree with the bulk of the remarks and will incorporate the suggested amendments in the manuscript. The Reviewer's questions are answered below (with references to the original submission).

**Q.** P. 2, line 15: The following sentence has no verb: "For typical thickness of IL of  $\leq 1$  m, the conductive heat flux amounts at  $\leq 1$  W m-2." Please rewrite

**A.** The verb was "amounts at". Rewritten as "is  $\leq 1 \text{ W m}^{-2}$ " for clarity.

**Q.** Page 3 – Section 2.1. If available, it would be useful to add more information on under-ice water climatology

**A.** Climatological data on water temperatures are unfortunately not available for the lake.

**Q.** In page 4, *I* is defined as net PAR irradiance, but in page 6 *I* is defined as the kinematic flux of the short-wave solar radiation. Please be consistent. In addition explain what you mean by short-wave solar radiation and which the relation between these and the PAR radiation. Please indicate the spectral bands of the definitions.

**A.** The spectrum of solar short-wave (wavelengths range 200-2500 nm) radiation is strongly modified by lake waters: clear water (or ice) quickly absorbs the long-wave (infrared) part of the spectrum and yellow substance absorbs the short-wave (ultraviolet) part. As a result, at < 1 m depth, > 95% of the penetrated radiation falls within the PAR spectral range of 400-700 nm (see e.g. Jerlov, 1976). In humic brown-water lakes, like Kilpisjärvi, the equivalence between PAR and total solar radiation is even closer (e.g. Leppäranta et al., 2010). Therefore, both terms are used interchangeably in the analysis. We clarify in the revised paper the relationship between the PAR and SW radiation flux. We also introduce different symbols for dynamic and kinematic radiation fluxes for clarity.

**Q.** Page 4, line 27: "sampling rate 0.5 Hz". This frequency is enough of to take in account all the relevant scales of the turbulence? Please comment.

**A.** The sampling rate is not critical for the velocity structure estimations of turbulence. The latter are calculated for every single "snapshot" along a corresponding acoustic beam and depend on spatial, not temporal resolution. The spatial resolution of 2 cm was apparently high enough to approach the mid- or short-wavenumber tail of the inertial interval (The Kolmogorov length scale is 0.3-1.0 cm for the dissipation rates of

 $10^{-10} - 10^{-8}$  W kg-1). The high sampling rate was simply chosen to provide a higher statistical significance of results at later averaging.

**Q.** Page 5 line 10: It is not clear how Noise is computed?

**A.** Eq. (5) is a line y = Ax + B, where Ax = r,  $y = C_v^{-3}D^{3/2}$ , B = Noise. The line was fitted by least-squares to the measured D(r). If Kolmogorov scaling is valid,  $B \equiv 0$ . Hence, *B* following from the LS-fit was used as an integral measure of side-effects (noise), and was applied as a quality-check parameter for the fitting. See also the works of Wiles et al. (2006) and McGinnis et al. (2015), cited in the paper.

**Q.** Page 5, line 11 "to discard corresponding values with subsequent interpolation between the neighboring values". I do not understand this sentence. Please explain better.

**A.** The text will be replaced with: "the measurements, for which Eq. 6 was valid, were abandoned as noisy."

**Q.** P5, line 14: "The estimations based on the averaging time of 20 min and r = 0.4 m were adopted for the further analysis 15 averaged over the three beam estimations.". Why?

**A.** The tested values r = 0.4, 0.5, 0.6 m and  $T_{averaging} = 10, 20, 30$  min were chosen as tentatively corresponding to the inertial range of turbulence, where the Kolmogorov scaling is valid (see also the reply to the question on Page 4, Line 27 above). All of them produced essentially the same output in terms of the TKE dissipation rate (see P. 5, Line 13), justifying the choice of the interval.  $T_{averaging} = 20$  min was adopted as the mid-value of the tested range; the lower value of the wavelength r = 0.4 m was adopted to avoid occasional influence on the results of the increased instrumental noise at higher distances from the acoustic head.

Q. Page 6 line 24: "warmer is the water". In the IL, right?

**A.** No. "The warmer is the CL" or "the higher is  $T_m$ " is correct; to be refined in the final

**version.**

**Q.** P. 6, line 23: The sentence "The ice was snow-free (snow thickness < 1 cm)" seems to be in contradiction to the information in line 20: "a thick ice-cover with more than 50% of snow-ice"

**A.** No contradiction. Snow-ice (or "white ice") is part of the ice cover consisting of the refrozen snow/water mix (see Kirillin et al., 2012, for details). The actual snow cover was practically absent. Text to be refined in the final version.

Q. P. 6. line 24: "a cross-section". The same cross-section used before?

A. practically the same.

**Q.** Page 8, line 14: "suggestion". Are there any evidence/reference for this suggestion?

**A.** No.

**Q.** Page 8, line 16: To integrate Eq. 3 the radiative profiles are not enough. How have you compute the other terms?

**A.** The other terms are calculated from measured temperatures as described in Section 2.2

**Q.** line 28: "The stratified IL occupied the 2-m thick layer". According to Figure 4 the IL is thinner than 2-m

A. changed to "up to 2 m"

**Q.** Page 9, Line 1: "the TKE dissipation rate  $\varepsilon$  increased". Increased relatively to what? Maybe the sentence should be rewritten beginning with the "background" values...

**A.** Increased relative to the previous values. As long as the time variations are discussed, the context is obvious.

**Q.** Page 9, Line 5: "The dataset on the TKE dissipation rate covered a longer period than the temperatures in the IL". How, as TKE dissipation rate is computed from the

temperatures?

A. Dissipation rates are not computed from temperatures. See Section 2.3 and Fig. 6.

**Q.** Page 9, line 14: " had no effect". Based on Fig. 7, it seems that there are some effect.

**A.** Some effect can indeed be speculated on. We remove the sentence as having no primary importance for the discussion.

Comments on the minor remarks:

The rest of remarks and suggestions, as referred to the figure order, figure legends, and term definitions are gratefully acknowledged and will be directly incorporated into the revised version of the paper.

**References**

Jerlov, N. G.

1976. *Marine optics*, Elsevier Oceanography Series 14. Amsterdam-Oxford-New York: Elsevier.

Kirillin, G., M. Leppäranta, A. Terzhevik, N. Granin, J. Bernhardt, C. Engelhardt, T. Efremova, S. Golosov, N. Palshin, P. Sherstyankin, et al.

2012. Physics of seasonally ice-covered lakes: a review. *Aquatic sciences*, 74(4):659–682. Leppäranta, M., A. Terzhevik, and K. Shirasawa

2010. Solar radiation and ice melting in Lake Vendyurskoe, Russian Karelia. *Hydrology Research*, 41(1):50–62.

McGinnis, D. F., G. Kirillin, K. W. Tang, S. Flury, P. Bodmer, C. Engelhardt, P. Casper, and H.-P. Grossart

2015. Enhancing surface methane fluxes from an oligotrophic lake: exploring the microbubble hypothesis. *Environmental science & technology*, 49(2):873–880.

Wiles, P. J., T. P. Rippeth, J. H. Simpson, and P. J. Hendricks 2006. A novel technique for measuring the rate of turbulent dissipation in the marine environment. *Geophysical Research Letters*, 33(21).